# Combinatorial Immunotherapies for Metastatic Colorectal Cancer

**DOI:** 10.3390/cancers12071875

**Published:** 2020-07-12

**Authors:** Eline Janssen, Beatriz Subtil, Fàtima de la Jara Ortiz, Henk M. W. Verheul, Daniele V. F. Tauriello

**Affiliations:** 1Department of Cell Biology, Radboud Institute for Molecular Life Sciences, Radboud University Medical Center, PO Box 9101, 6500 HB Nijmegen, The Netherlands; Eline.Janssen@radboudumc.nl (E.J.); Beatriz.Subtil@radboudumc.nl (B.S.); Fatima.delaJaraOrtiz@radboudumc.nl (F.d.l.J.O.); 2Department of Medical Oncology, Radboud University Medical Center, PO Box 9101, 6500 HBNijmegen, The Netherlands; Henk.Verheul@radboudumc.nl

**Keywords:** metastasis, checkpoint blockade, TGF-β, immune suppression, solid tumour, tumor microenvironment

## Abstract

Colorectal cancer (CRC) is one of the most frequent and deadly forms of cancer. About half of patients are affected by metastasis, with the cancer spreading to e.g., liver, lungs or the peritoneum. The majority of these patients cannot be cured despite steady advances in treatment options. Immunotherapies are currently not widely applicable for this disease, yet show potential in preclinical models and clinical translation. The tumour microenvironment (TME) has emerged as a key factor in CRC metastasis, including by means of immune evasion—forming a major barrier to effective immuno-oncology. Several approaches are in development that aim to overcome the immunosuppressive environment and boost anti-tumour immunity. Among them are vaccination strategies, cellular transplantation therapies, and targeted treatments. Given the complexity of the system, we argue for rational design of combinatorial therapies and consider the implications of precision medicine in this context.

## 1. Introduction

The intestine is a vital organ for food digestion and has an important barrier function. This comes with an intricately balanced immune system that acts against pathogens, while tolerating beneficial microbiota as well as the foreign molecule antigens present in our food. Furthermore, immune cells can also detect and eliminate transformed cells to prevent the development of tumours, a process that is called immunosurveillance [1,2]. Successful cancers escape this mechanism in ways that are poorly understood [3]. Moreover, inflammatory signalling has also been linked to the formation and progression of tumours [4,5,6]. Although the underlying biology of cancer metastasis still has many aspects that remain incompletely explained, recent decades have seen a lot of progress—including in the field of cancer immunology. Unfortunately, colorectal cancer (CRC) has so far been mostly refractory to immunotherapies in the clinic. Nevertheless, accumulating evidence suggests that this will change. In this review, we give an immuno-oncological overview for this disease and describe immunotherapeutic strategies currently under development.

### 1.1. Tumorigenesis and Disease Progression

CRC is an adenocarcinoma that originates in epithelial cells of the large intestine. These epithelial cells are organized as a single-cell layer lining the inner surface of the intestinal tube, and are surrounded by a complex stroma that consists of supportive tissue, immune infiltrates, blood- and lymph vessels, neurons, and muscles. Different routes have been described that contribute to the transformation of normal intestinal epithelium into a tumour, reviewed in more detail elsewhere [7]. The majority of CRCs arise from spontaneous mutations, frequently leading to activation of the WNT/β-catenin pathway [8]. Resulting polyps or adenomas can accumulate further alterations that transform them into more aggressive adenocarcinomas. These events frequently include driver mutations in the mitogen-activated protein kinase (MAPK) and bone morphogenic protein (BMP)/transforming growth factor-β (TGF-β) pathways—as well as in tumour suppressor gene *TP53* [9,10,11,12]. Besides this classical version of colorectal tumorigenesis there is also the serrated pathway, with precursor lesions differing on histological architecture as well as molecular characteristics [13]. Serrated tumours can become deficient in DNA mismatch repair, which can lead to hypermutated CRCs that also acquire atypical numbers of tandem repeats [14]. These cancers are also called microsatellite instable (MSI) tumours, a portion of which arise from hereditary mutations in DNA mismatch repair genes (Lynch syndrome) [15]. In contrast to hypermutated/MSI tumours, CRCs that are microsatellite stable (MSS) typically accumulate moderately low numbers of mutations [16].

As carcinomas become more invasive, they can migrate into the vasculature and spread to distant sites in the body. About half of the patients that are diagnosed with localized CRC already have cancer cells in one or more distant organs, albeit still undetectable [17]. Indeed, genetic evidence suggests that cancer dissemination may be an early event [18,19]. Months to years after surgical removal of the primary CRC, these cells can cause disease recurrence. Whereas primary CRC can often be completely removed by surgery, metastases are often more difficult to treat. Consequently, most deaths are due to (extensive) metastatic CRC (mCRC), the main focus of this review. Although multiple organs can be affected, including lungs, peritoneal cavity, bones, and brain; liver metastasis is the most common and best-studied form. Nevertheless, many questions about this process remain unanswered [20,21].

### 1.2. Tumour Heterogeneity

Besides the genetic background, many additional parameters are taken into consideration for disease prognosis. In current CRC staging practice, these include histopathological observations such as differentiation grades, cellular phenotypes, tumour budding, and lymph node involvement—many of which have been formalized in the TNM (tumour, lymph node, metastasis) classification. These parameters correlate with both disease outcome and metastatic patterns [22], indicating biological relevance. Another clinical parameter that is linked to disease outcome is the primary tumour location: ascending and transverse colon (right), versus descending and sigmoid colon (left) [23]. Despite all these factors, predicting a patient’s risk of metastasis is still a challenge.

To further dissect tumoral heterogeneity and explore new treatable targets, extensive molecular classification attempts have been made. Aside from the abovementioned frequent driver mutations, there is considerable genetic variation between tumours—without clearly ascribed prognostic value. This prompted a shift in focus, and technology, towards gene expression. A number of large transcriptomic stratification efforts have been reported, consolidated in a system with four consensus molecular subtypes (CMS), of which CMS4 has the worst prognosis [24]. Although this classification has not yet substantially impacted on clinical practice, it uncovered new biological aspects of CRC.

### 1.3. Focus on the Tumour Microenvironment

In parallel with transcriptomic studies that were mostly focused on epithelial cancer cells, an additional paradigm emerged in understanding disease progression: a complicit tumour microenvironment (TME), or tumour stroma. The TME consists of the cellular components surrounding the mutated cancer cells (i.e., tissue parenchymal cells, fibroblasts, immune infiltrates and vascular cells), as well as signalling molecules and metabolites, physical conditions (e.g., pH, oxygen, stiffness), and other factors such as the microbiota [4,25,26,27]. This marked complexity has long precluded in-depth analysis of the role of the TME in tumour progression and metastasis. However, specific and context-dependent roles of the TME in harbouring or advancing metastatic lesions have emerged.

For example, cancer-associated fibroblasts (CAFs) are recognized as a main constituent of tumours and have heterogeneous phenotypes, including paracrine functions that drive tumour progression [28,29]. Relatedly, TGF-β, a key activating growth factor for fibroblasts, was found to correlate with poor prognosis [30,31,32]. Specifically, levels of both ligand mRNA (*TGFB1*, −2 and −3) and downstream target gene expression in CAFs—and other cell types of the TME—carry robust prognostic value [31,33]. In fact, much of the predictive power that is captured in the CMS4 is associated with the TME [33,34]. Nevertheless, epithelial CRC cell-intrinsic signatures have also been found to facilitate patient stratification [35,36]. In about half of CRCs, epithelial TGF-β signalling is disrupted by mutations, suggesting that—in cancer progression—this pathway mainly acts on the TME. However, this may discount the role of TGF-β in CRCs that do not have such mutations. Interestingly, in serrated adenomas, TGF-β induces a mesenchymal phenotype—echoing with the well-described function of this cytokine in activating an epithelial-to-mesenchymal transition [37]—and contributing to the TGF-β-high, poor prognosis CMS4 CRCs [38,39].

Other influential components of the TME are cell types of the immune system, including antigen presenting cells (APCs) and cytotoxic immune cells. Of the first group, mainly macrophages and dendritic cells (DCs) have been studied in the context of CRC. In general, their presence is associated with anti-tumour immune responses, although their activation is commonly inhibited by the TME— reversing the balance towards immune suppression. This malignant polarization is best-described for tumour-associated macrophages (TAMs), which can use immunomodulatory cytokines, metabolism, and checkpoint molecules to regulate the anti-tumour immune response [40]; as well as for myeloid-derived suppressor cells (MDSCs), a heterogeneous collection of relatively immature immune cells with potent suppressive activities [41]. Conversely, cytotoxic cell types—T cells and natural killer (NK) cells—are important for immunosurveillance and may block metastasis initiation. Cytotoxic T lymphocytes (CTLs) can be powerful, long-lived cancer cell killers, yet are activated to a single specific peptide. NK cells are less specific, but they can still recognize cancer cells. Furthermore, NK cells also kill cancer cells through antibody-dependent cellular toxicity (ADCC), which is relevant for a number of targeted therapies.

Cytotoxic immune cells activation requires concerted action by other immune cell types and is very sensitive to negative regulation. Successful tumours exploit the richness of negative regulatory mechanisms of the immune system to thwart immunosurveillance [27]. Clearly, this immune contexture is relevant for cancer diagnosis, prognosis, and treatment decisions [42]. Importantly, the density and type of infiltrating T cells is a good prognostic marker [43], and has been formalized and validated in the Immunoscore [44,45]. The understanding of cancer as an ecosystem has amplified research into therapeutic strategies that target the whole tumour, rather than the currently predominant focus on the epithelial cancer cells [46].

## 2. Clinical Practice: Systemic Chemotherapy-Based Treatment of mCRC

Surgical removal of tumours remains the main treatment option for CRC, both in the setting of primary disease and, increasingly, also for patients with oligometastases—a clinical diagnosis that is generally defined as less than five detectable metastatic lesions in maximally two organs [47]. Improvements in the survival of patients undergoing resection of liver metastases from CRC have been reported in mostly observational studies [48,49,50]. In general, these reports indicate that curation can be obtained when complete resection is feasible. Alternative local treatment strategies are radiofrequency, microwave ablation or stereotactic ablative radiotherapy [51,52,53,54,55,56]. These may also improve survival as long as the disease is localized in liver or lungs. In patients with more extensive disease, local interventions have not provided survival benefit when only a part of the metastases can be removed [57,58]. Alternatively, complete debulking of multiple metastatic lesions may improve survival and is subject of an ongoing study when combined with systemic treatment [59] (NCT01792934)—results are expected in 2022/2023.

When local treatment alone fails or is not feasible, several systemic treatment options are available. Before the era of chemotherapy, patients with mCRC had a median overall survival of 6 months [60]. With the introduction of the chemotherapeutic agents fluoropyrimidines (5-FU or capecitabine), oxaliplatin and irinotecan—and later on the targeted agents against vascular endothelial growth factor (VEGF) and epidermal growth factor receptor (EGFR) in combination therapy and/or sequential administration—overall survival has improved to over 30 months in the most recent randomized phase III trials [61,62,63]. Multiple combination treatment strategies with these cytotoxic agents have been developed and studied, resulting in established combinations of 5-FU with oxaliplatin or irinotecan [64,65,66]. The antiangiogenic monoclonal antibody against VEGF, bevacizumab, is not active by itself but has some benefit in combination with chemotherapy added to either of the combinations mentioned above [67], and this may be related to a specific genetic alteration [68]. Recently, triple therapy with the drug regimen FOLFOXIRI (fluorouracil, leucovorin, oxaliplatin, and irinotecan) with bevacizumab has been evaluated in depth as first line therapy in relatively young patients with a good performance status [69]. Although some benefit in progression-free survival and potential overall survival was found compared to therapy with 5-FU, oxaliplatin and bevacizumab alone in the control group, this combination entailed increased toxicity.

The anti-EGFR monoclonal antibodies cetuximab and panitumumab are active by themselves, resulting in a survival benefit with adequate selection of patients with left-sided tumours with a *RAS* wild-type status [70]. Other targeted therapies that were developed for a specific subgroup of patients include treatment with trastuzumab/pertuzumab for HER2+ mCRCs [71], and the combination of encorafenib (BRAF inhibitor) and cetuximab—which was demonstrated to be efficacious and approved by regulatory agencies for *BRAF V600E*-mutant mCRC [72]. Additional trials investigating regorafenib (a multikinase inhibitor) and trifluridine/piperacil (a nucleoside analogue) have shown modest improvement compared to best supportive care [73,74]. Multiple other targeted agents, especially tyrosine kinase inhibitors (TKIs), have failed in mCRC for unknown reasons. Today, none of the systemic therapies provide a realistic chance for curation in patients with mCRC.

## 3. Immunotherapies

In recent decades, immunotherapies for cancer have seen a renaissance. A landmark development was the discovery and subsequent therapeutic blockade of immunological checkpoint molecules cytotoxic T-lymphocyte-associated protein-4 (CTLA-4) and PD-L1, a ligand to programmed death receptor-1 (PD-1) [75]. Antibodies against these cell surface proteins, known as immune checkpoint inhibitors (ICIs), have been applied with remarkable responses in advanced melanoma. This outcome has been expanded to a number of other cancer types, including advanced non-small cell lung cancer, metastatic urothelial carcinoma, advanced renal cell carcinoma, hepatocellular carcinoma, head and neck squamous cell carcinoma, and recurrent lymphoma [76].

Antibodies against PD-1 were also tested in patients with mCRC [77]. Although promising responses were seen in MSI tumours, this was not the case for MSS cancers —the type that the vast majority of patients with mCRC have. This ostensible lack of success is likely associated with low tumour mutational burden and, therefore, with a low number of tumour-specific neo-antigens, which is also reflected in the observed scarcity of tumour-infiltrating lymphocytes (TILs) in many MSS cancers [78]. However, full immune activation is a complex process that involves both innate and adaptive immune systems [79], and may encounter TME interference on various levels in this multistep immune cycle. Consequently, disparate treatment types are being designed and investigated, each aiming to close the circle. These can be divided in three main topics: boosting immune recognition to stimulate effector responses; circumventing in situ immune regulation by using active immune cells, and overcoming immunosuppressive signalling. Subsequently, we will discuss immuno-oncology combinations that aim to overcome mono-therapeutic shortcomings.

### 3.1. Boosting Immune Recognition

Already before much of the complex regulation of (anticancer) immunity was understood, Dr. William Coley found that intratumoral injection of bacterial extracts could induce curative effects—placing the first known instance of cancer immunotherapy at the end of the 19th century [80]. Modern knowledge of these signalling mechanisms offers the possibility of a more controlled approach to therapeutically boost cancer immunity. This can be achieved by stimulating pathogen- or damage-associated molecular pattern (PAMP/DAMP) recognition signalling, mediated by toll-like receptors (TLRs) and stimulator of interferon genes (STING) in innate immune cell types (Figure 1). A number of molecules have been designed to stimulate these mechanisms [81,82], which elicit the secretion of pro-inflammatory cytokines. This response aids the maturation of APCs such as DCs, as well as their ability to present tumour-associated antigens (TAAs) to T cells [83]. Although individual efforts have shown potential as monotherapies in mCRC [84,85] (Table 1), these approaches seem not be sufficient to elicit powerful immune responses that can eliminate tumours.

Nevertheless, these agents also serve as immunoadjuvants in vaccination approaches [86,87], which can be applied either in early-stage disease—to prevent relapse—or in overt mCRC, as a method to boost immune responses that may control or reject advanced disease (Figure 1). One option is the use of autologous tumour cells, inactivated with e.g., radiation. Preclinical studies showed that such vaccinations can be effective in inhibiting tumour growth and extending mouse survival, linked to increased lymphocyte infiltration and responses [88,89,90,91]. Three phase III clinical trials of patients with stage II/III CRC investigated the intradermal use of irradiated autologous tumour cells, using Bacillus Calmette-Guerin (BCG) as an immunoadjuvant, demonstrating a decrease in risk of recurrence [92]. This result was not reached in a small randomized trial conducted after complete resection of metastases (stage IV), assessing the efficacy of irradiated metastases-derived CRC cells incubated with Newcastle disease virus—although some survival benefits were found [93].

In a somewhat related approach, patients are injected with attenuated viruses that selectively infect and kill cancer cells. Besides increasing the detectability of tumour antigens upon cell rupture, the immunogenic cancer cell death induced by these oncolytic viruses releases PAMPs and DAMPs, boosting APC maturation and adaptive immune responses [94] (Figure 1). Studies in preclinical CRC models with a mutated herpes simplex virus (G207) and an adenovirus (Ad881) demonstrated cancer regression [95,96]. In the clinic, this strategy is met with early successes, but also with challenges such as optimal systemic delivery [97,98,99,100] (Table 1).

Viruses have also been used as vectors to express TAAs inside the patient. In fact, several vaccination strategies are being evaluated to elicit or boost specific immune responses in situ [112,113]. While MSI/hypermutated CRC is already associated with a high number of potential neo-antigens, vaccination is seen as an additional possibility in raising immunogenicity, especially in Lynch syndrome for some relatively frequent TAAs [114]. For MSS CRC, several specific antigens are under consideration, although none of these have a very high prevalence [115]. Preclinical studies vaccinating against the *MYB* proto-oncogene showed suppressed tumour growth through induction of T-cell-mediated anti-tumour immunity [116,117]. Similar results were obtained for viral vaccination strategies targeting *GUCY2C* [118,119], prompting a successful phase I clinical trial (Table 1). There has also been progress using mRNA-encoded vaccines [120]. A proof-of-concept study demonstrated the feasibility of harnessing cancer-genomics to synthesize personalized poly-neo-epitope mRNAs that conferred anti-tumour immunity in mouse models, including of CRC [121,122]. Clinical translation of this idea was recently reported for metastatic melanoma [123]. In addition, a number of generic or personalized peptide vaccines have shown potential in the preclinical setting [124,125,126,127] and in the clinic (Table 1).

Rather than activating APCs in situ, an alternative strategy is the administration of ex vivo stimulated and activated dendritic cells (Figure 1). For this approach, DCs are isolated from the patient, activated, pulsed with tumour lysates or specific TAAs, and then reinfused—often in combination with an adjuvant. DC vaccines have been studied in multiple clinical trials [109,110,111,128,129,130,131]. Although generally demonstrating safety—and in most cases increased tumour-specific T-cell concentrations [109,110,129,130,131]—therapeutic benefit is reported in only a subset of trials [104,110,111] (Table 1). Related to DC cell therapy is the concept of administrating synthetic immune niches; transplantable 3D-scaffolds that locally provide chemoattractants, tumour antigens, and adjuvants to recruit and activate DCs and T cells [132,133]. Although not implemented specifically for CRC, preclinical studies on lymphoma [134] and melanoma [135,136,137] show promising effects.

If priming tumour-reactive T cells is not the key problem, absent or inadequate stimulatory signalling cues from the TME can be counterbalanced therapeutically. Several antibodies have been designed to promote CTL activity (Figure 1). Agonistic targeting of the co-stimulatory receptors OX40 (CD134), 4-1BB (CD137), and glucocorticoid-induced tumour necrosis factor receptor-related protein (GITR) can increase infiltration and activity of effector T cells while decreasing infiltration of regulatory immune cells in preclinical CRC models [138,139,140,141,142]. These antibodies have entered clinical investigation (Table 1). Furthermore, a recent study reports the adaptation of an autoimmunity-directed CD40 inhibitor into an agonist by isotype switching, resulting in anti-cancer activity [143]. In an alternative strategy, bispecific T-cell engagers (BiTEs) have been designed to physically summon T lymphocytes to CRC cells; this proximity alone can in some cases be sufficient to elicit cytotoxic effects [144]. These agents are antibody derivatives with one arm binding CD3 on T cells and the other a cell surface protein on CRC cells, such as EpCAM [145,146], EGFR [147], CEA [148,149], and glycoprotein A33 [150]. Some of these therapies have entered clinical trials (Table 1).

### 3.2. Circumventing In Situ Immune Activation: Adoptive Cell Therapy

Beyond providing immune-boosting signals or administering pre-activated DCs, it is also possible to bypass all in situ immune priming and inject patients with activated cytotoxic immune cell products (Figure 2). In the 1960s and 70s, seminal studies with adoptive cell therapy (ACT) of autologous TILs helped establish the concept that T cells can recognize and kill tumour cells [151]. This strategy demonstrated encouraging clinical responses in patients with metastatic melanoma. Progress in other tumours including CRC has been complicated by the relative underrepresentation of tumour-reactive T cells in these cancers, although efforts remain ongoing (Table 2).

T cells are highly dependent on human leukocyte antigen (HLA) matching—the variability of which precludes broad, off-the-shelf functionality. As an alternative to the need for autologous and antigen-specified immune cells, NKs have also been used as a therapeutic agent [152]. Interestingly, the use of allogenic NK products—sourced from healthy blood donors, umbilical cord blood or even cell lines—may have several benefits over autologous cells [153,154]. Moreover, preclinical efforts indicated that ACT of peripheral blood NK cells can synergize with antibody-based targeted therapy such as cetuximab via ADCC—independently of *RAS*/*RAF* mutation status [155,156,157]. Two phase I trials affirmed safety and suggested biological activity of this combination in patients with gastrointestinal cancers (Table 2).

The HLA-unrestricted killing ability that characterizes NK cells could be a great advantage when transferred to T cells, inspiring a number of engineered T-cell receptor (TCR) approaches (Figure 2). For instance, the chimeric antigen receptor (CAR)—a fusion of an immunoglobulin epitope-recognition domain with signalling regions of the TCR and co-stimulatory co-receptors—may be an auspicious therapeutic option for solid tumours with a moderate or low tumour mutational burden [161]. Although this treatment requires antigens to be expressed on the target surface, these antigens do not have to be presented by the HLA machinery, which is sometimes downregulated in tumour cells [162]. The most extensively studied CAR-target in CRC is CEA. This strategy showed anti-tumour responses in preclinical models [163,164,165] and is under evaluation in the clinic (Table 2). The targets of additional CARs that demonstrated promising in vivo results in CRC models—and that are now in clinical trials—include EpCAM [166,167], GUCY2C [168,169], NKG2D [170], and HER2 [171] (Table 2). Others such as doublecortin-like kinase 1 (DCLK1)—a proposed cancer stem cell marker—show preclinical potential [172]. Moreover, CAR-T cells might be exploited to target CAFs: T cells with a CAR against fibroblast activating protein (FAP) significantly inhibited subcutaneous growth of various solid tumours [173]. However, efficacy for CRC was not convincingly demonstrated, likely due to the strong immunosuppressive TME associated with CRC. Indeed, this problem may generally limit survival of (CAR) T cells [174,175]. Of note, significant toxicity has been linked to CAR-T-cell therapy [176]. Potential strategies relieving toxicity, and likely increasing efficacy, could involve localized administration of CAR-T cells [165,177,178].

Generally appreciated for their better safety profile, NK cells have also been genetically modified to express CARs. The NK-92 cell line was engineered to express a CAR targeting CEA [179] as well as EpCAM [180]. Adoptive transfer of either inhibited growth of subcutaneous human CRC xenografts, in mice. Furthermore, intraperitoneal infusion of allogeneic NKG2D CAR-NK cells reduced tumour burden in one patient with CRC liver metastasis [181]. In further attempts to pair the innate cytotoxic powers of NK cells with specific recognition ability, NK-92 cells have been equipped with a functional TCR; conferring phenotypic traits of T cells, while NK cell effector functions were retained [182]. An alternative approach that would be HLA-unrestricted is a recently cloned TCR, which—by binding the invariable HLA-relative MR1—can recognize various cancer cell types [183]. Approaches like this may be another step towards safe, efficacious, and off-the-shelf targeted immunotherapies. Nevertheless, clinical translation of ACT for solid tumours like CRC is still in early development.

### 3.3. Targeting Immunosuppressive Signalling

Similar to immune-boosting strategies, the efficacy of immune cell therapy is severely limited by the TME (Figure 3). Overcoming this immunosuppressive milieu could therefore be vital in allowing durable clinical responses in patients with advanced CRC [46]. This concept was recently illustrated with the inhibition of stromal TGF-β signalling [184,185]. TGF-β, besides activating CAFs, is a potent immune suppressor [186,187] and was therefore hypothesized to have an active role in immune evasion—putatively explaining its link to poor prognosis. Upon the generation of a genetic and transplantable, immunocompetent metastatic mouse model for CMS4-like MSS CRC, this hypothesis was tested. Indeed, the key role of TGF-β in metastatic initiation is the suppression of anti-tumour immune responses: inhibition of the pathway efficiently blocked metastatic liver colonization in a T-cell-dependent manner [184]. There are several other immunomodulatory cytokines that are associated with disease progression in CRC [188,189]; among them are IL-6 and IL-33 [190,191,192] (Figure 3). In mice, IL-6 deficiency reduced liver metastasis, concomitant with increased activity of DCs and cytotoxic T cells [193], and IL-33 facilitated metastasis by increasing myeloid cell infiltration and neo-angiogenesis [194]. Interestingly, administration of a soluble IL-33 receptor—which is downregulated in mCRC—normalized the TME and suppressed metastasis in mice [195].

Furthermore, chemokines and their receptors regulate the recruitment of immune cells to the TME and thereby constitute interesting immunotherapeutic targets [196] (Figure 3). Preclinically, chemokine (receptor) inhibitors have been shown to suppress CRC liver metastasis formation and inhibit tumour growth by reducing the accumulation of immature myeloid cells and CAFs [197,198,199,200,201]. For instance, a CXCR4 antagonist suppressed metastasis by decreasing TME infiltration of CAFs and MDSCs [198]. Clinical data support the premise that chemokines help shape the immunosuppressive landscape, highlighting inflammatory crosstalk mechanisms between myeloid cells and T lymphocytes at the invasive margin of CRC liver metastases [201,202]. A pro-tumorigenic signalling cascade, mediated by CCL5, could be therapeutically interrupted by CCR5 antagonists. This therapy elicited therapeutic responses in a phase I study, including the repolarization and redistribution of TAMs, and the subsequent activation of T cells [201]. Regarding TAMs, a considerable fundamental and (pre-)clinical effort has been dedicated to targeting them [203,204], including a focus on inhibiting colony-stimulating factor 1 (CSF-1, also known as M-CSF) or its receptor CSF-1R [205]. An anti-CSF-R1 antibody was found to deplete TAMs in a murine CRC model as well as in patients, which associated with increased lymphocyte infiltration and slower tumour growth [206]. Despite this, as well as proving safe, no objective response was observed [207].

As metabolic reprogramming in the TME can also be a driver of immune evasion, immunometabolism has emerged as a new therapeutic frontier [208]. One example is the essential amino acid tryptophan: important for T-cell activation and catabolized by IDO1 (indoleamine 2,3-dioxygenase) into a suppressive immunomodulator (Figure 3). Expression of this enzyme is elevated in CRCs that have few TILs and in patients with poor prognosis [209,210,211]. Preclinical studies with IDO1 inhibitors in CRC models confirmed its role as an immune suppressor, but indicated that this strategy may fall short by itself [212,213]. Accordingly, early clinical trials could not show efficacy of a selective small-molecule IDO1 inhibitor (epacadostat) as monotherapy for solid malignancies [214]. Nevertheless, a recent study in mice targeting IDO1 with a short hairpin RNA found robust innate immune responses impeding tumour growth—outperforming epacadostat [215]. Furthermore, new ICI strategies have yielded good results in preclinical models of CRC (Figure 3), including the blockade of inhibitory receptors such as LAG-3 and TIM-3 [216,217,218,219,220]. Finally, the depletion of regulatory T cells may also contribute to negating tumour-induced immunosuppression [221].

### 3.4. Combinatorial Immuno-Oncology

It is becoming clear that few if any monotherapies will achieve broad, durable efficacy for advanced (solid) cancers, such as mCRC. Consequently, as seen in the development of chemotherapies, treatments like ICIs are increasingly used in pairs or together with other agents [222,223] (Table 3). For MSI mCRC, the combination of anti-CTLA-4 and anti-PD-1 proved quite successful in terms of response rate and survival outcomes [224]. Interestingly, a recent early stage neo-adjuvant trial with the same treatment combination demonstrated promising immune responses in primary tumours—even for MSS CRC—which might, in follow-up studies, translate to lower recurrence rates [225]. Furthermore, ongoing phase I trials for mCRC test the same two antibodies with a TLR9 agonist together with radiosurgery, or an anti-PD-1 antibody with a STING agonist (Table 3).

A strong dependency of effective immune checkpoint blockade on correctly functioning DCs has been reported [226,227], and a substantial replenishment of new tumour-reactive T-cell clones has been observed following PD-1 inhibition [228,229]. These findings indicate a potential synergy between ICI and vaccination therapy. This rationale is supported by preclinical studies, such as the combination of the *MYB* vaccine and anti-PD1 therapy [117], which has now entered a phase I trial (Table 3). Similarly, a KRAS peptide vaccine in combination with anti-PD-1 and anti-CTLA-4 is being investigated in patients with advanced CRC (Table 3). Somewhat relatedly, a specific KRAS G12C-mutant inhibitor was demonstrated to have a surprising immunotherapeutic effect by itself and to synergize with ICIs in a preclinical model [230], and has progressed to clinical testing (Table 3).

Some vaccination approaches have shown stronger efficiency when combined with chemotherapeutic agents [232,233,234], and the depletion of regulatory T cells by low-dose cyclophosphamide treatment offered a survival benefit to patients with mCRC that were treated with modified vaccinia Ankara-5T4 vaccination [235]. Also, there are clinical studies investigating oncolytic viruses in combination with ICI (Table 3). An added potential for oncolytic viruses is the engineered encoding of cytokines, antigens, inhibitors or other biological agents—rendering them into multiplexed immunomodulatory platforms [236]. Furthermore, combinations have been proposed of oncolytic viruses with TGF-β inhibition [237] or with CAR-T cells (Table 3). Interestingly, the latter may serve as viral carriers, addressing the currently limiting factor of oncolytic virus delivery [238].

Agonistic T-cell-directed antibodies are also being combined with other types of immunotherapy, e.g., CEA-CD3 BiTEs together with PD-1/PD-L1 blockade [239,240] (Table 3). Moreover, anti-OX40 treatment has been tested together with different agonistic agents [241] or with anti-PD-1 in preclinical models [242], and these strategies are being translated into the clinic. Following the observation of limited clinical therapeutic benefit of a GITR agonist as a monotherapy [100], a phase I/II study has been initiated to test this agent in combination with ICIs (Table 3). Similarly, the combination of antibodies stimulating CD27—another important co-stimulatory receptor of T cells [243]—with anti-PD-1 therapy is clinically evaluated (Table 3). Interestingly, agonistic anti-4-1BB antibodies were reported to increase the efficacy of CAR-T cells, not just by stimulating T-cell activity, but also by reducing the levels of infiltrating regulatory T cells and MDSCs [244]—a goal that could also be achieved by inhibiting IDO1 [245] or STAT3 [246]. Additionally, antibodies against CD25 (targeting regulatory T cells) and CSF-1R (targeting TAMs or MDSCs) have also been co-administered with ICIs in preclinical models to give promising results [247,248] (Table 3).

Regarding ACT, ICI combinations with TIL transfusion therapy are being tested in the clinic (Table 3). The anti-tumour activity of CEA CAR-T cells was increased by PD-L1 blockade in preclinical models; again involving MDSCs as a major source of checkpoint molecules [165]. Furthermore, a recent preclinical study demonstrated superior anti-tumour responses of anti-CD30/CEA bispecific CAR-T cells over CEA monospecific CAR-T cells. The additional anti-CD30 modality was proposed to antagonize negative regulation of effector T cells and might potentiate CAR-T-cell therapies in a broad manner [249]. Other combinations with CAR-T cell therapy include p38 and PKD inhibitors, which can stabilize interferon receptor IFNAR1 to increase T cell persistence in the TME—as observed with a FAP-CAR-T approach in preclinical models of CRC [174].

In the aforementioned metastatic mouse model for MSS mCRC, in which TGF-β inhibition could counteract the immunosuppressive TME and prevent metastatic initiation [184], this treatment was not effective for established liver metastases. Likewise, the monotherapy of with anti-PD-L1 antibodies resulted in minimal benefit. However, dual blockade led to potent, curative T-cell-mediated immune responses [184], confirming that immunotherapies can be effective upon overcoming (TGF-β-induced) immune evasion. Similar conclusions were drawn from other studies, including in MSI CRC or different cancer types, promulgating high levels of stromal TGF-β signalling as a therapeutic target as well as a prognostic and predictive biomarker in the context of (ICI) immunotherapy [250,251,252,253,254,255,256,257,258,259]. Not only T-cell-based approaches benefit from targeting the immunosuppressive TME, as TGF-β inhibition significantly increased infiltration and cytotoxicity of adoptively transferred NK cells [260]. Indeed, strategies inhibiting TGF-β signalling may play a key role in a range of effective immunotherapeutic combinations [184,186,250,251,252] and several inhibitors are in (early) clinical trials, including in combination with ICIs (Table 3). Other signalling inhibitor plus ICI combinations in clinical evaluation include a CCR2/CCR5 antagonist (Table 3). Interestingly, CCR2 antagonists [261], and other compounds targeting myeloid cells [262], might be particularly effective in combination with ICI for patients that underwent radiotherapy. Furthermore, epacadostat is being evaluated in combination with ICI as well as with the epigenetic agent azacytidine (Table 3), and a preclinical study reported a synergy of dual blockade of IDO1 and TLR7 [263].

Many of these studies illustrate the concept that clinically advanced immunotherapies are expected to be significantly potentiated by auxiliary treatments that overcome the immunosuppressive TME. The aim for these combinations is often the increase of immune cell infiltration and activation, or involves the depletion, re-education or repolarization of Tregs, CAFs, TAMs, or MDSCs [264]. New concepts and combinations that promise to meet this ambition continue to come out of preclinical research, such as ICI plus TNF blockade [265], ICI with BRAF and MEK inhibitors [266,267], or a combination of 4 agents designed to engage a variety of innate and adaptive immune cells [268]. Thus, with the expansion of both our understanding of cancer immunity and our therapeutic arsenal, concerted intervention at several critical points in the immune cycle is an emergent paradigm.

Although most studies for combinatorial immunotherapies described in this section are in early stages, some early results temper expectations. For example, the phase I study of the combination of anti-PD-1 with CCR5 inhibitor showed limited clinical activity, with an objective response rate (ORR) of only 5% [231]. Moreover, two clinical failures concerning patients with mCRC—one a phase II study of the addition of anti-PD-L1 to a combination of fluoropyrimidine with bevacizumab in maintenance after first line therapy [269], the other a phase III trial with a MEK inhibitor plus anti-PD-L1 in the third line [270]—indicate substantial challenges to translating promising preclinical activities [264]. As with some of the clinical disappointments in targeted therapies, the lack of predictive biomarkers for patient selection may contribute to negative trials.

## 4. Future Perspectives

Beyond the uncertainties involved in the clinical validation of new treatment strategies and combinations, determining the right combination among an ever-growing set of possibilities will become an even bigger challenge for immunotherapies than it already is for more conventional therapies. This is due to the heterogeneity between patients, as well as the high level of complexity within the tumours: to our limited understanding of the interactions between all components of the TME and of the underlying immunosuppressive signalling mechanisms. Improved insight could substantially prioritize approaches, reducing the number of relevant combinatorial possibilities. Recent and ongoing high-dimensional approaches on clinical samples with a strong focus on the TME, including single-cell techniques, are anticipated to help towards that goal. Moreover, these observations should be paired with improved experimental models. Until recently, preclinical models of CRC—typically involving heterotopic (i.e., subcutaneous) and/or xenotypic injections of (human) cell lines into mice—often recapitulated neither tumour heterogeneity nor the TME, likely contributing to failure in clinical translation. However, more sophisticated alternatives are being developed that allow the study of key effectors and molecular signalling networks in sufficient complexity, facilitate the design and validation of combinatorial treatments, and may enable the identification of biomarkers. In our view, suitable models would have to be either faithful, immunocompetent metastatic mouse models [184,271,272,273], or in vitro cultured TME models such as those based on patient-derived organoids or tumour explants [274,275]—potentially in combination with humanized mouse models.

Furthermore, adequate tools are needed to stratify patients based on their TME [276]. General progress is being made, such as a pan-cancer immunogenomics study that distinguished six immune subtypes [277]; and recent work on subtyping patient with CRC, based on TME [278,279,280], may make clinical testing of new therapies more efficient. For example, (combinatorial) treatment strategies designed for a given TME subtype—that potentially could be present in different cancer types—would benefit from basket trials [281]. Within this trend fits the tissue-agnostic FDA approval of anti-PD-1 antibodies in 2017. This ICI approval for patients with advanced mismatch-repair-deficient solid tumours, such as MSI-mCRC, was based on trials with in total 149 patients spanning 15 cancer types [282]. Also, further validation of immune subtypes—especially from primary CRC material—will improve prognostic resolution, and may even pave the way for a more individualized disease outlook [42,283]. In this scenario, an immunological balance, or set point, within the individual TME would be assessed to determine the best countermeasures [284]. Consequently, treatment combinations may increasingly become biomarker-driven and personalized, and their efficacy may be predicted by bespoke patient-derived TME culture models or humanized xenograft avatars.

An additional challenge for combinatorial immunotherapy lies in optimizing the timing and delivery sequence of different therapies, affecting both efficacy and toxicity. Preclinical studies addressing these factors indicate the significance of appropriate scheduling for combinations of immunotherapies, e.g., with radiotherapy [242,285,286,287]. Furthermore, as immunotherapies continue to gain ground for mCRC, the positioning of such treatment modalities in the context of disease management might need to be evaluated. Conventional wisdom would suggest second- or third-line implementation of experimental immuno-oncology strategies for patients with metastatic disease. However, perhaps presurgical, neo-adjuvant immunotherapy—with tumour and -draining lymph nodes still in place—may give a better result; as unleashing a strong immune response in the primary tumour could immunize the patient against future metastases [225,288,289]. A prudent approach would be to determine the long-term benefits of such therapies in high-risk patients first.

## 5. Conclusions

Despite the initial failure of immune checkpoint inhibitors in MSS mCRC, a host of promising immunotherapeutic options are under development that may be instrumental in extensively improving survival for patients with advanced disease. The realization that—like in many solid tumours—the CRC microenvironment strongly suppresses anti-tumour immune responses has prompted the development of strategies that include stromal modulation. As the complexity of interactions within the TME is steadily being dissected, we propose that an ecological type of cancer therapy will be attainable in the near future. This outlook promises precision immuno-oncology with individually optimized treatment combinations, and gives hope to more curative therapies for patients with metastatic CRC.

## Figures and Tables

**Figure 1 cancers-12-01875-f001:**
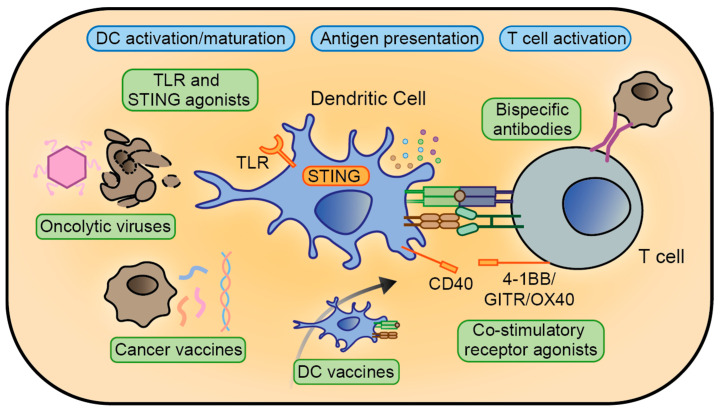
Schematic overview of therapeutic strategies aimed at boosting immune recognition. This process involves dendritic cell (DC) activation and maturation, antigen processing and presentation to T cells, and the subsequent activation of T cells with cytokines and other co-stimulatory signals. Some of these steps can be circumvented using ex vivo-generated DC vaccines or by the application of bispecific antibody products. Cancer cells are depicted in grey.

**Figure 2 cancers-12-01875-f002:**
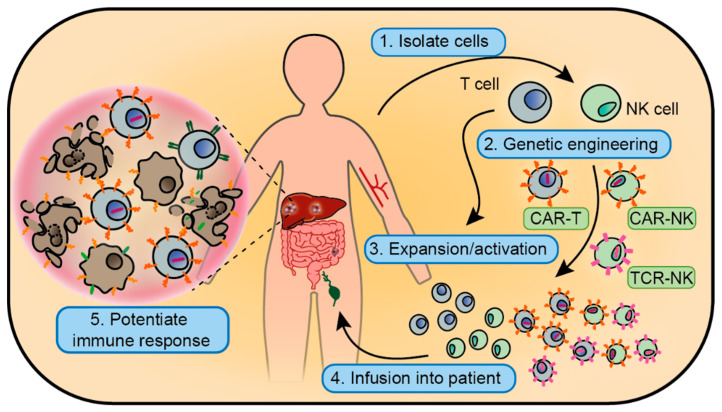
Schematic overview of adoptive cell therapies. T cells or NK cells can be isolated from the patient, or from healthy donors, and expanded ex vivo. During this process, they can be genetically engineered to express a CAR or TCR variant. Upon infusion, these killer cells can reinforce the existing immune response or effect a new one.

**Figure 3 cancers-12-01875-f003:**
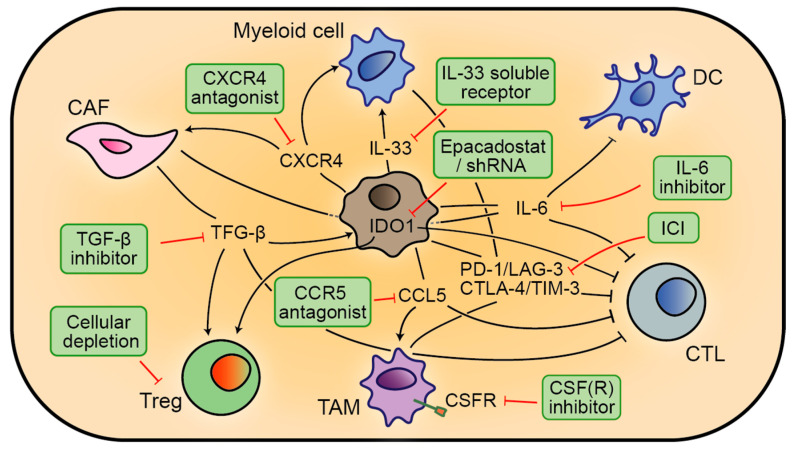
Schematic overview of some of the players, targets, and strategies that are involved in modulating and re-educating the immunosuppressive TME for mCRC, with the aim of promoting anti-tumour immunity. An IDO1-expressing epithelial CRC cell (grey) is depicted in the middle.

**Table 1 cancers-12-01875-t001:** Therapeutic approaches to boost immune recognition in clinical trials for mCRC.

Type	Agent	Stage	Reference
*Boosting Agents*			
Small molecules	TLR9 agonist (MGN1703)	Phase II	[84]
	TLR9 agonist (MGN1703)	Phase III	NCT02077868
	STING agonist (EE7766)	Phase I	NCT04144140
Antibodies/BiTEs	OX40 agonist (MEDI6469) + liver metastasis ablation	Phase I	NCT02559024
	Anti-4-11B agonist (urelumab)	Phase I/II	[101]
	Anti-GITR agonist (AMG228)	Phase I	[102]
	CEA-CD3 bispecific antibody (cibitasamab)	Phase I	NCT02324257
	CEA-CD3 bispecific antibody (MEDI-565)	Phase I	NCT01284231
	CEA-CD3 bispecific antibody (AMG 211)	Phase I	NCT02291614
	EpCAM-CD3 bispecific antibody (solitomab)	Phase I	[103]
*Vaccination Approaches*			
Autologous tumour cells	Irradiated autologous tumour cells combined with BCG (OncoVAX^®^) (intradermal vaccine)	Phase III	[92]
	Irradiated metastases-derived tumour cells incubated with Newcastle disease virus	Phase II/III	[93]
	OncoVAX^®^ and surgery	Phase III	NCT02448173
	Autologous or Allogeneic tumour cells	Phase I/II	NCT00722228
Oncolytic viruses	Genetically altered herpes simplex virus (NV1020)	Phase I/II	[97]
	Enadenotucirev	Phase I/II	[98]
Peptide vaccine	Ad5-GUCY2C-PADRE	Phase I	[104]
	SART3 peptide vaccine	Phase I	[105]
	hCG peptide vaccine conjugated to diphtheria toxoid	Phase II	[106]
	13-peptide cocktail vaccine	Phase I/II	[107]
	Personalized peptide vaccine	Phase II	[108]
	Personalized peptide vaccine	Phase I	NCT02600949
DC vaccine	CEA or Frameshifted peptide-loaded DCs (Lynch syndrome/MSI)	Phase I/II	NCT01885702
	Ex vivo CD40L-activated DCs	Randomized	[109]
	CEA pulsed DC + tetanus toxoid and IL-2	Phase I	[110]
	Autologous tumour antigens-loaded DC	Phase II	[111]
	Autologous tumour lysate activated DC	Phase II	NCT02919644

**Table 2 cancers-12-01875-t002:** Clinical trials investigating immune cell therapies in mCRC.

Cell Therapy	Intervention/Target	Stage	Reference
TILs	Lymphodepletion + autologous TIL + IL-2	Phase II	NCT03610490
	Lymphodepletion + autologous TIL + IL-2	Phase II	NCT01174121
	Lymphodepletion + autologous TIL + IL-2	Phase II	NCT03935893
NK cells	Allogenic NK + cetuximab	Phase I	[158]
	Adoptive transfer NK + trastuzumab/cetuximab-based chemotherapy	Phase I	[159]
CAR-T cells	Anti-EGFR-IL12 CAR-T Cells	Phase I/II	NCT03542799
	Anti-EGFR CAR-T cells	Phase I/II	NCT03152435
	Anti-MUC1 CAR-T cells	Phase I/II	NCT02617134
	Anti-EpCAM CAR-T cells	Phase I/II	NCT03013712
	Anti-CEA CAR-T cells	Phase I	[160]
	Anti-CEA CAR-T cells	Phase I	NCT02850536
	Anti-CEA CAR-T cells	Phase I	NCT02349724
	Anti-EGFR CAR-T cells	Phase I/II	NCT01869166
	Anti-NKG2D CAR-T cells	Phase I	NCT03370198
	Anti-NKG2DL CAR-(γδ) T cells	Phase I	NCT04107142
	Anti-DR5 CAR-T cells	Phase I/II	NCT03638206
CAR-NK cells	Anti-MUC1 CAR-NK cells	Phase I/II	NCT02839954

**Table 3 cancers-12-01875-t003:** Active clinical trials investigating combinatorial immunotherapies in mCRC.

Combinatorial Strategy					Stage	Ref.
TLR9 agonist	+	radiosurgery	+	anti-PD1 & anti-CTLA4	Phase I	NCT03507699
STING agonist (MK-1454)			+	anti-PD1	Phase I	NCT03010176
*MYB* vaccine			+	anti-PD1	Phase I	NCT03287427
KRAS peptide vaccine			+	anti-PD1 & anti-CTLA4	Phase I	NCT04117087
Pexa-Vec oncolytic virus			+	anti-PD1 & anti-CTLA4	Phase I/II	NCT03206073
Oncolytic adenovirus (enadenotucirev)			+	anti-PD-1	Phase I	NCT02636036
Anti-CEA-CD3 bispecific antibody			+	anti-PD1	Phase I	NCT02650713
Anti-GITR			+	anti-PD1 & anti-CTLA4	Phase I/II	NCT03126110
Anti-CD27			+	anti-PD1	Phase I/II	NCT02335918
Anti-HER2 CAR-T cells			+	oncolytic adenovirus	Phase I	NCT03740256
Autologous TIL			+	anti-PD1	Phase II	NCT01174121
Autologous TIL			+	chemotherapy	Phase II	NCT03935893
Anti-PD1 activated autologous TIL			+	chemotherapy	Phase I/II	NCT03904537
Anti-NKG2D CAR-T cells			+	chemotherapy	Phase I	NCT03692429
Anti-CEA CAR-T cells			+	chemotherapy	Phase I/II	NCT02959151
Anti-NKG-2 CAR-T cells			+	chemotherapy	Phase I	NCT03310008
Anti-CEA CAR-T cells			+	internal radiation therapy	Phase I	NCT02416466
TGF-β inhibitor			+	anti-PD1	Phase I/II	NCT03724851
TGF-βRII (extracellular domain; ligand trap)			fused to	anti-PD-L1	Phase I/II	NCT03436563
CCR5 inhibitor			+	anti-PD1	Phase I	NCT03274804 [231]
CCR2/CCR5 inhibitor			+	anti-PD1 or chemotherapy	Phase I/II	NCT03184870
Fluoropyrimidine and bevacizumab			+	anti-PD-L1	Phase III	NCT02291289
CSF-1R inhibitor			+	anti-PD-L1	Phase I	NCT02777710
IDO1 inhibitor + azacytidine			+	anti-PD1	Phase I/II	NCT02959437
K-RAS(G12C) inhibitor			+	anti-PD1	Phase I/II	NCT03600883
Anti-TIM-3 antibody			+	anti-PD-1	Phase I	NCT02817633
Anti-TIM-3 antibody			+	anti-PD-1	Phase I	NCT03099109
Anti-LAG-3			+	anti-PD-1	Phase I/II	NCT01968109
Anti-LAG-3			+	anti-PD-1	Phase I	NCT03250832
Anti-LAG-3			+	anti-PD-1	Phase I	NCT03005782
Anti-LAG-3			+	anti-PD-1	Phase I	NCT03219268

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
