# Peer review of "Combinatorial Immunotherapies for Metastatic Colorectal Cancer"

_cancers, 2020, doi:10.3390/cancers12071875_

Round 1

Reviewer 1 Report

I have only minor comments

  1. Point 4.1 can be substituted as a specific point because this is in fact an introduction of the other 3 major points. In fact, the second paragraph that specifically evaluate TGF-B can be included in the area of targeting immunosuppressive signaling (I suggest that this would be the point 4.3)
  2. I suggest that 4.5 Combinatorial immuno-oncology, would be probably integrated inside the 3 main blocks. That would help to clarify current failures and potential improvements.
  3. It has been published (at least in part in different symposium) negative results in some of the new potential combinations. For instance, results of PICASSO trial with the combo of pembrolizumab and CCR5 inhibitor has been presented in the ASCO 2020 Meeting (ORR 1/20). In addition, the combination of galunisertib (TGFb inh) plus durvalumab shows also very modest efficacy in pancreatic cancer (ORR 1/42). These negative trials and other potentially negative trials should be mentioned to remark that probably in unselected patients, combination therapies can also provide limited activity. In fact, recently 2 phase III trials, with the combination of a MEK inhibitor plus atezolizumab vs regorafenib and the combination of bevacizumab plus atezolizumab vs bevacizumab failed in third line and maintenance after first line therapy, in mCRC.
  4. In the area 2.1 Tumorigenesis and disease progression at the end of the last paragraph…liver or lungs, local therapy such as surgery or radiotherapy…should be probably substituted by ….liver or lungs, local therapy such as surgery, stereotactic body radiation or radiofrequency…..
  5. At the end of point 3 (last paragraph). (73). Just recently, triple therapy…..FDA and EMA approved the combination of encorafenib plus cetuximab but not the triplet therapy. This should probably be changed. In addition, there is a mistake with the BRAF mutation (V600 instead of V6700).

Author Response

We thank the reviewer for his/her valuation of this manuscript as well as the critical and supportive comments. As to the specific (minor) comments:

  1. We have removed the subheading that was 4.1. Furthermore, we have split the paragraph on TGF-beta into two parts, the monotherapy and the combination therapy, and moved them to their respective places with the rest of the treatments as suggested.
  2. We do see the point: that many combinations are logically linked to failures of monotherapies and might therefore also be expected to be mentioned together. However, for the breadth of therapeutic concepts, integrating the combinations with the 3 pillars of immunotherapy would soon become quite disorganized, in our experience. In a separate sub-section, we have at least been able to maintain some level of structure. To communicate our choice, and perhaps curtail the confusion mentioned by reviewer 1, we state the separation of monotherapies and combinations up-front, in the introductory text preceding the individual I/O blocks.
  3. This is an excellent point, and we have added the negative CRC-specific results with our interpretation: that there are plenty of challenges ahead and not all combination results will be positive, despite great preclinical rationale. We also include the point of patient selection.
    This new paragraph is located at the end of the combinations section (now §3.4). With respect to the examples given by the reviewer, we found that the last trial (bevacizumab plus atezolizumab vs bevacizumab in maintenance after first line; MODUL - NCT02291289) is a phase II instead of a phase III.
  4. We found the mentioned phrase in the (old) section 2.1 to be overlapping with what is also mentioned in the ‘clinical practice’ section (old section 3, now 2). Rather than correcting it, we decided to remove the former sentence (in the introduction).
  5. We have found (and changed to) the more recent reference for the results of that trial, and also found the FDA approval notice of the double therapy. This has been corrected, along with the BRAF V600E issue.

Reviewer 2 Report

The authors describe the complexity of tumour microenvironment (TME) in colorectal cancer and its potential predictive role with regard to benefit from immune checkpoint inhibitors and provide an extensive literature review of trials assessing combination strategies to overcome immunotherapy resistance.

The topic of this paper is a pressing and yet unmet need in the treatment of metastatic colorectal cancer.

Taking into account the aim of this review, the authors should consider that the introduction is too long and this could divert attention from the principal purpose of the paper. Moreover, the paragraph “3. Clinical practice: systemic chemotherapy-based treatment of mCRC” falls outside the context of the review. It could be eliminated.

Although the immune components influencing TME are described with exhaustive clarity, the authors places little emphasis on myeloid-derived suppressor cells (MDSCs) and their potent immune suppressive activity.

The author gives more space to preclinical model and phase I studies and this approach is very appreciable and innovative, while very far from being applied to clinical practice.

Although the paper is well written and minutely discussed, some points should be addressed:

  • There are two paragraphs called 2.2.
  • With regard to TME, paragraphs 2.2 (Focus on the tumour microenvironment) and 2.3 (Cancer immunity) could be condensed into a single paragraph.
  • In paragraph “3. Clinical practice: systemic chemotherapy-based treatment of mCRC”,
  • at line 168, I consider excessive the word “tremendous” because FOLFOXIRi plus bevacizumab is considered a feasible regimen;
  • line 176: for BRAF V600-mutant mCRC, the potential new option is the doublet (approved by FDA).
  • The development of strategies aimed at TME modulation is the principal focus of this review. This should emerge in the title that is currently too general.

Author Response

We thank the reviewer for his/her time, and for the sharp comments that have—we believe—improved our manuscript. As the reviewer appreciates, this review aims to include relevant concepts from both basic research and clinical translation. We stand by our decision to combine this viewpoint with an overview of the clinical reality, and therefore choose not to eliminate the latter section (number 3 in the old version, number 2 in the new). We argue that there are parallel challenges and lessons between conventional therapy development and immuno-oncology that we acknowledge throughout the text. To better accentuate these parallels, we have made some adjustments in this section, as well as corrections related to recent updates from clinical trials—as helpfully commented by reviewers 1 and 2.

The corollary of our approach is that introducing these concepts threatens to become lengthy, as the reviewer correctly remarked. To streamline this section, and curtail attention being diverted unduly, we have restructured sections 1+2 into a single section, removed 12 lines in total (1/4 page), and eliminated as many distractions we could find.

Furthermore, we have placed more emphasis on MDSCs by adding a sentence to the introduction. We note that there were already 5 passages in which MDSCs were mentioned in the main text.

As to the remaining points:

  • We have adjusted the sub-headings and sub-sections, as indicated
  • We have removed the charged word ‘tremendous’
  • We have corrected the doublet therapy for BRAF mutant mCRC
  • Title: Although we agree that TME modulation can be seen as a principle focus of this review, we interpret that the reviewer defines this as equivalent to ‘overcoming immune suppression’. Our vision of this manuscript is broader, and we have clearly devoted attention to improving immune recognition and adoptive cell therapies (the other two main blocks), to end up endorsing rational combinations of, or with, immunotherapies. This broad premise justifies a broad title.

Reviewer 3 Report

The manuscript entitled “Combinatorial immunotherapies for metastatic colorectal cancer“ represents a well-written, comprehensive, well-arranged overview of the topic based on a thorough review of the literature. It should be published in “Cancers” after a minor revision.

I have a very few comments on the text.

  1. A number of claims are based on results of pre-clinical studies. Therefore I would find it useful to extend the relevant part of the text (paragraph 5: Future perspectives, line 465 et seq.) with a more detailed description of the animal models which are currently used. As well, some description of that which look promising as they could answer the still unresolved questions would be beneficial.
  2. There are two points which I do not consider to be precisely expressed. First, neutrophils are not commonly listed as cytotoxic cells (lines 121-122). The role of neutrophils in cancer is multifactorial and not fully understood. It can be context and tumor dependent, and in many instances, neutrophils promote the tumor growth and spreading. Indeed, neutrophils can produce substance that have the capacity to damage bystander cells, and some studies have shown that neutrophils can e.g. antagonize the metastatic spreading, such as in lung carcinoma. The therapeutic targeting of neutrophils rather focuses on strategies to avoid the deleterious effects of neutrophils in cancer and to reduce their activity instead of increasing their activity to fight cancers.
  3. The second point is the use of dendritic cell vaccines based on precursors from healthy (thus allogeneic) donors. These DCs may enhance the T cell anti-cancer immune responses in recipients by other mechanisms, rather than by direct antigen presentation to T cells due to potential HLA incompatibility. These points should be briefly explained in the text.

Author Response

We thank the reviewer for the praise and appreciation of both the contents and structure of our manuscript.

With respect to the comments:

  • Agreeing it is very useful to talk about models, we have nevertheless had to resist the urge of adding a lengthy separate section on this, given the already strainingly broad premise.
    As we already described the relevant concepts (and ensuing model types) that can move the field forwards (highlighted in the text), we did see the need to add a concise sentence of what is our issue with models as they have been used up until recently: that they are often heterotopic and/or xenotypic, i.e. mostly (human) CRC cell lines subcutaneously injected in mice; and that these are key reasons for the model recapitulating neither heterogeneity nor the (correct) TME.
  • Neutrophils: because of the multifactorial nature of their functions in cancer as explained by the reviewer, as well as their absence from the rest of this review, we decided the easiest solution is to remove them altogether.
  • Similarly—although we take the point of the reviewer—we actually found no example of allogeneic DC use in the therapies that we mention. The easiest solution was to remove the mention of healthy-donor DCs.

We thank the reviewer for pointing out these issues.